# Boosting Transferability of Targeted Adversarial Examples via Hierarchical Generative Networks

**Xiao Yang**[1]  **Yinpeng Dong**[1]  **Tianyu Pang**[1]

## Abstract

Transfer-based adversarial attacks can effectively evaluate model robustness in the black-box setting. Though several methods have demonstrated impressive transferability of untargeted adversarial examples, targeted adversarial transferability is still challenging. In this paper, we develop a simple yet practical framework to efficiently craft targeted transfer-based adversarial examples. Specifically, we propose a conditional generative attacking model, which can generate the adversarial examples targeted at different classes by simply altering the class embedding and share a single backbone. Extensive experiments demonstrate that our method improves the success rates of targeted black-box attacks by a significant margin over the existing methods — it reaches an average success rate of 29.6% against six diverse models based only on one substitute white-box model in the standard testing of NeurIPS 2017 competition, which outperforms the state-of-the-art gradient-based attack methods (with an average success rate of $<2\%$) by a large margin. Moreover, the proposed method is also more efficient beyond an order of magnitude than gradient-based methods.

## 1. Introduction

Recent progress in adversarial machine learning demonstrates that deep neural networks (DNNs) are highly vulnerable to adversarial examples (Szegedy et al., 2014; Goodfellow et al., 2015), which are maliciously generated to mislead a model to produce incorrect predictions. It has been demonstrated that adversarial examples possess an intriguing property of transferability (Liu et al., 2017; Wu et al., 2020; Huang et al., 2019; Demontis et al., 2019; Yang

[1]Department of Computer Science and Technology, Institute for AI, THBI Lab, Tsinghua University, Beijing 100084, China. Correspondence to: Xiao Yang <yangxiao19@mails.tsinghua.edu.cn>.

*Accepted by the ICML 2021 workshop on A Blessing in Disguise: The Prospects and Perils of Adversarial Machine Learning.* Copyright 2021 by the author(s).

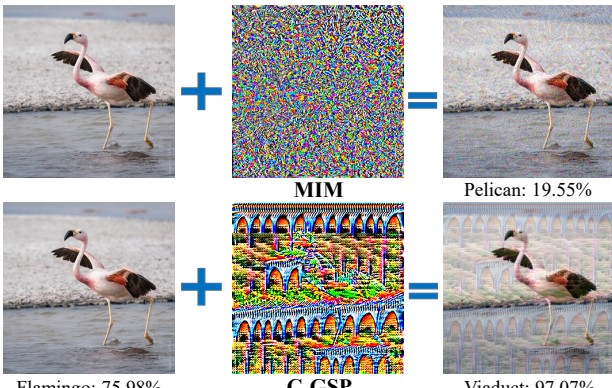

*Figure 1.* The targeted adversarial examples crafted by MIM (Dong et al., 2018) and the conditional generative semantic pattern (C-GSP) crafted by our method for the Inception-v3 (Szegedy et al., 2016) model given the target class *Viaduct* with perturbation budget 16 under the $\ell_\infty$ norm constraint. We also show the predicted labels and probabilities of these images by the black-box model DenseNet-201 (Huang et al., 2017).

et al., 2020) — the adversarial examples crafted for a white-box model can also mislead other unknown models, making *black-box attacks* feasible. The current methods have achieved impressive performance of untargeted black-box attacks, intending to cause misclassification of the black-box models (Liu et al., 2017; Dong et al., 2018). However, the targeted *black-box attacks*, aiming at misleading the black-box models by outputting the adversary-desired target class, perform unsatisfactorily (Dong et al., 2020) and have not been extensively explored (Zhang et al., 2020).

Existing efforts on targeted black-box attacks can be categorized as instance-specific and instance-agnostic attacks. Specifically, the instance-specific attack methods (Goodfellow et al., 2016; Moosavi-Dezfooli et al., 2016; Kurakin et al., 2017; Dong et al., 2018) craft adversarial examples by performing gradient updates iteratively, which achieve unsatisfactory performance for targeted black-box attacks due to easy overfitting to a white-box model (Dong et al., 2018; Xie et al., 2019). On the other hand, the instance-agnostic attack methods learn a universal adversarial perturbation (Zhang et al., 2020) or a universal function (Song et al., 2018; Naseer et al., 2019) on the data distribution independent of specific instances. They can promote more general and transferable adversarial examples since the universal pertur-

bation or function can alleviate the data-specific overfitting problem by training on an unlabeled dataset. Represent CD-AP (Naseer et al., 2019) adopts a generative model as a universal function to obtain an acceptable performance when facing one specified target class. However, CD-AP needs to learn a generative model for each target class while performing multi-target attack (Han et al., 2019). Thus it is not scalable to the increasing number of targets such as hundreds of classes, limiting practical efficiency.

To address the aforementioned issues and develop an effective targeted black-box attack, in this paper we propose a conditional generative model as the universal adversarial function to craft adversarial perturbations. Thus we can craft adversarial perturbations targeted at different classes, using a single model backbone with different class embeddings. The proposed generative method is simple yet practical to obtain superior performance of targeted black-box attacks, meanwhile with two technical improvements including *smooth projection mechanism* that better helps the generator to probe targeted semantic knowledge from the classifier and *adaptive Gaussian Smoothing* with the focus of making generated results obtain adaptive ability against adversarially trained models. Thus ours only trains one model and reaches an average success rate of 51.1% against six naturally trained models and 36.4% against three adversarially trained models based only on one substitute white-box model in NeurIPS ImageNet dataset, which outperforms CD-AP by a large margin of 6.0% and 31.3%, respectively.

While handling plenty of classes (*e.g.*, 1,000 classes in ImageNet), the effectiveness of generating targeted adversarial examples will be affected by a single generative model due to the difficulty of loss convergence in adversarial learning (Xu et al., 2019; Berthelot et al., 2017). Thus we train a feasible number of models (*e.g.*, 10∼20 models on ImageNet) to further promote the effectiveness beyond the single model backbone. Specifically, each model is learned from a subset of classes specified by a designed hierarchical partition mechanism by considering the diversity property among subsets, for seeking a balance between effectiveness and scalability. It reaches an average success rate of 29.6% against six different models, outperforming the state-of-the-art methods with an average success rate of <2% by a large margin, based only on one substitute white-box model in the NeurIPS 2017 competition. Moreover, the proposed method achieves substantial speedup over gradient-based methods.

Furthermore, these adversarial perturbations generated by the proposed Conditional Generative models can arise as a result of strong Semantic Pattern (C-GSP) as shown in Fig. 1. We experimentally find that the generated adversarial semantic pattern itself achieves well-generalizing performance among the different models and is robust to the influence of data, which is very instructive for the understanding of adversarial examples.

## 2. Method

In this section, we introduce a conditional generative model to learn a universal adversarial function, which can achieve effective multi-target black-box attacks. While handing plenty of classes, we design a hierarchical partition mechanism to make the generative model capable of specifying any target class under a feasible number of models, regarding both the effectiveness and scalability.

### 2.1. Problem Formulation

We use $\boldsymbol{x}_s$ to denote an input image belonging to an unlabeled training set $\mathcal{X}_s \subset \mathbb{R}^d$, and use $c \in \mathcal{C}$ to denote a specific target class. Let $\mathcal{F}_\phi : \mathcal{X}_s \to \mathbb{R}^K$ denote a classification network that outputs a class probability vector with $K$ classes. To craft a targeted adversarial example $\boldsymbol{x}_s^*$ from a real example $\boldsymbol{x}_s$, the targeted attack aims to fool the classifier $\mathcal{F}_\phi$ by outputting a specific label $c$ as $\arg\max_{i \in \mathcal{C}} \mathcal{F}_\phi(\boldsymbol{x}_s^*)_i = c$, meanwhile the $\ell_\infty$ norm of the adversarial perturbation is required to no more than value $\epsilon$ as $\|\boldsymbol{x}_s^* - \boldsymbol{x}_s\|_\infty \leq \epsilon$.

Although some generative methods (Poursaeed et al., 2018; Naseer et al., 2019) can learn targeted adversarial perturbation, it does not take into account the effectiveness of multi-target generation, thus leading to inconvenience. To make the generative model learn how to specify multiple targets, we propose a conditional generative network $\mathcal{G}_\theta$ that effectively crafts multi-target adversarial perturbations by modeling class-conditional distribution, as illustrated in Fig. 2. The conditional generative model $\mathcal{G}_\theta : (\mathcal{X}_s, \mathcal{C}) \to \mathcal{P}$ learns a perturbation $\boldsymbol{\delta} = \mathcal{G}_\theta(\boldsymbol{x}_s, c) \in \mathcal{P} \subset \mathbb{R}^d$ on the training data. The output $\boldsymbol{\delta}$ of $\mathcal{G}_\theta$ is projected within the fixed $\ell_\infty$ norm, thus generating the perturbed image $\boldsymbol{x}_s^* = \boldsymbol{x}_s + \boldsymbol{\delta}$.

Given a pretrained network $\mathcal{F}_\phi$, we propose to generate the targeted adversarial perturbations by solving

$$
\begin{aligned}
\min_\theta &\mathbb{E}_{(\boldsymbol{x}_s \sim \mathcal{X}_s, c \sim \mathcal{C})}[\mathbb{CE}(\mathcal{F}_\phi(\mathcal{G}_\theta(\boldsymbol{x}_s, c) + \boldsymbol{x}_s), c)], \\
&\text{s.t. } \|\mathcal{G}_\theta(\boldsymbol{x}_s, c)\|_\infty \leq \epsilon.
\end{aligned}
\tag{1}
$$

By solving problem (1), we can obtain a targeted conditional generator by minimizing the loss of specific target class in the unlabeled training dataset. Note that we only optimize the parameter $\theta$ of the generator $\mathcal{G}_\theta$ using the training data $\mathcal{X}_s$, then the targeted adversarial example $\boldsymbol{x}_t^*$ can be crafted by $\boldsymbol{x}_t^* = \boldsymbol{x}_t + \mathcal{G}_\theta(\boldsymbol{x}_t, c)$ for any given image $\boldsymbol{x}_t$ in the test data $\mathcal{X}_t$, which only requires an inference for this targeted image $\boldsymbol{x}_t$. We experimentally find that the objective (1) can enforce the transferability for the generated perturbation $\boldsymbol{\delta}$. A reasonable explanation is that $\boldsymbol{\delta}$ can arise as a result of **strong** and **well-generalizing** *semantic pattern* inherent to the target class, which is robust to the influence of any training data.

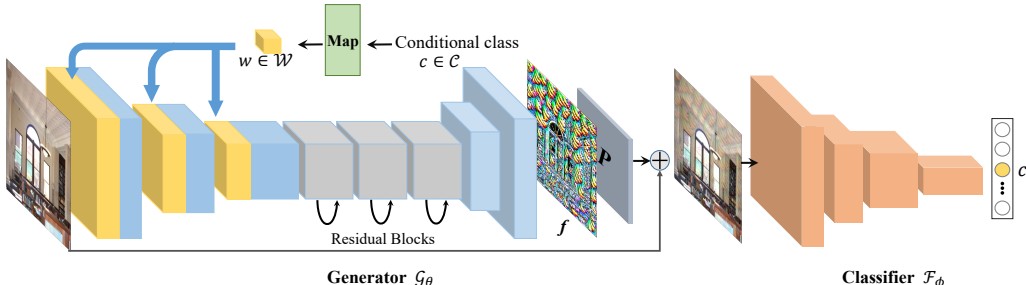

*Figure 2.* An overview of our proposed generative method for crafting C-GSP, which includes modules of conditional generator and classifier. The generator integrates the image and conditional class vector from Map network into a hidden incorporation. Note that only the generator is trained in the whole pipeline to probe the target boundaries of the classifier.

## 2.2. Network Architecture

We now present the details of the conditional generative model for targeted attack, as illustrated in Fig. 2. Specifically, we design a mapping network to generate a target-specific vector in the implicit space of each target and train conditional generator $\mathcal{G}_\theta$ to reflect this vector by constantly misleading the classifier $\mathcal{F}_\phi$.

**Mapping network.** Given an one-hot class encoding $\mathbb{1}_c \in \mathbb{R}^K$ from target class $c$, the mapping network aims to generate the targeted latent vector $\boldsymbol{w} = \mathcal{W}(\mathbb{1}_c)$, where $\boldsymbol{w} \in \mathbb{R}^M$ and $\mathcal{W}(\cdot)$ consists of a multi-layer perceptron (MLP) and a normalization layer, which can construct diverse targeted vectors $\boldsymbol{w}$ for a given target class $c$. Thus $\mathcal{W}$ is capable of learning effective targeted latent vectors by randomly sampling different classes $c \in \mathcal{C}$ in training phase.

**Generator.** Given an input image $\boldsymbol{x}_s$, the encoder first calculates the feature map $\boldsymbol{F} \in \mathbb{R}^{N \times H \times W}$, where $N$, $H$ and $W$ refer to the number of channels, height and width of the feature map, respectively. The target latent vector $\boldsymbol{w}$, derived from the mapping network $\mathcal{W}$ by introducing a specific target class $c$, is expanded along height and width directions to obtain the label feature map $\boldsymbol{w}_s \in \mathbb{R}^{M \times H \times W}$. Then the above two feature maps are concatenated along the channels to obtain $\boldsymbol{F}' \in \mathbb{R}^{(N+M) \times H \times W}$. The obtained mixed feature map is then fed to the subsequent network. Thus our generator $\mathcal{G}_\theta$ translates an input image $\boldsymbol{x}_s$ and latent target vector $\boldsymbol{w}$ into an output image $\mathcal{G}_\theta(\boldsymbol{x}_s, \boldsymbol{w})$, which enables $\mathcal{G}_\theta$ to synthesize adversarial images of a series of targets. For the output of feature map $\boldsymbol{f} \in \mathbb{R}^d$ in the decoder, we adopt a **smooth projection** $P(\cdot)$ to perform a change of variables over $\boldsymbol{f}$ rather than directly minimizing its $\ell_2$ norm as (Han et al., 2019) or clipping values outside the fixed norm (Naseer et al., 2019), which can be denoted as

$$\boldsymbol{\delta} = P(\boldsymbol{f}) = \epsilon \cdot \tanh(\boldsymbol{f}), \qquad (2)$$

where $\epsilon$ is the strength of perturbation. Since $-1 \leq \tanh(\boldsymbol{f}) \leq 1$, $\boldsymbol{\delta}$ can automatically satisfy the $\ell_\infty$-ball bound with perturbation budget $\epsilon$. This transformation can be regarded as a better smoothing of gradient than directly clipping values outside the fixed norm, which is also in-

strumental for $\mathcal{G}_\theta$ to probe and learn the targeted semantic knowledge from $\mathcal{F}_\phi$.

**Training objectives.** The training objectives seek to minimize the classification error on the perturbed image of the generator as

$$\theta^* \leftarrow \arg\min_\theta \mathbb{CE}\Big(F_\phi\big(\boldsymbol{x}_s + \mathcal{G}_\theta(\boldsymbol{x}_s, \mathcal{W}(\mathbb{1}_c))\big), c\Big), \quad (3)$$

which adopts an end-to-end training paradigm with the goal of generating adversarial images to mislead the classifier the target label, and $\mathbb{CE}$ is the cross entropy loss.

## 2.3. Hierarchical Partition for Classes

While handling plenty of classes, the effectiveness of a conditional generative model will decrease because the representative capacity is limited with a single generator. Therefore, we propose to divide all classes into a feasible number of subsets to train models when the class number $K$ is large, *e.g.*, 1,000 classes in ImageNet. To obtain a good partition, we introduce a representative target class space, which is nearly equivalent to the original class space $\mathcal{C}$. Specifically, we utilize the weights $\phi_{cls} \in \mathbb{R}^{D \times C}$ in the classifier layer for the classification network $\mathcal{F}_\phi$. Therefore, $\phi_{cls}$ can be regarded as the alternative class space since the weight vector $\boldsymbol{d}_c \in \mathbb{R}^D$ from $\phi_{cls}$ can represent a class center of the feature embeddings of input images with same class $c$.

To capture more diverse examples in a given sampling space, we adopt K-determinantal point processes (DPP) (Kulesza & Taskar, 2012; 2011) to achieve a hierarchical partition, which can take advantage of the diversity property among subsets by assigning subset probabilities proportional to determinants of a kernel matrix. First, we compute the RBF kernel matrix $L$ of $\phi_{cls}$ and eigendecomposition of $L$, and a random subset $V$ of the eigenvectors is chosen by regarding the eigenvalues as sampling probability. Second, we select a new class $c_i$ to add to the set and update $V$ in a manner that de-emphaseizes items similar to the one selected. Each successive point is selected and $V$ is updated by Gram-Schmidt orthogonalization, and the distribution shifts to avoid points near those already chosen.

*Table 1.* Transferability comparison for multi-target attacks on ImageNet NeurIPS validation set (1k images) with the perturbation budget of $\ell_\infty \leq 16$. The results are averaged on 8 different target classes. Note that CD-AP[†] indicates that training **8 models** can obtain results, while our method only train **one** conditional generative model. * indicates white-box attacks.

| Method | Time (ms) | Models | Naturally Trained | | | | | | | Adversarially Trained | | |
|---|---|---|---|---|---|---|---|---|---|---|---|---|
| | | | Inc-v3 | Inc-v4 | IncRes-v2 | Res-152 | DenseNet | GoogleNet | VGG-16 | Inc-v3$_{ens3}$ | Inc-v3$_{ens4}$ | IncRes-v2$_{ens}$ |
| MIM | ∼130 | - | 99.9* | 0.8 | 1.0 | 0.4 | 0.2 | 0.2 | 0.3 | <0.1 | 0.1 | < 0.1 |
| TI-MIM | ∼130 | - | 98.5* | 0.5 | 0.5 | 0.3 | 0.2 | 0.4 | 0.4 | 0.3 | 0.3 | 0.2 |
| SI-MIM | ∼130 | - | 99.8* | 1.5 | 2.0 | 0.8 | 0.7 | 0.7 | 0.5 | 0.3 | 0.3 | 0.1 |
| DIM | ∼130 | - | 77.0* | 2.7 | 0.5 | 0.8 | 1.1 | 0.4 | 0.8 | 0.1 | 0.2 | 0.1 |
| TI-DIM | ∼130 | - | 52.5* | 1.1 | 1.2 | 0.5 | 0.5 | 0.5 | 0.8 | 0.4 | 0.6 | 0.4 |
| SI-DIM | ∼130 | - | 90.2* | 3.8 | 4.4 | 2.0 | 2.2 | 1.7 | 1.4 | 0.5 | 0.5 | 0.2 |
| CD-AP[†] | ∼15 | 8 | 94.2* | 57.6 | 60.1 | 37.1 | 41.6 | 32.3 | 41.7 | 1.5 | 2.2 | 1.2 |
| CD-AP-gs[†] | ∼15 | 8 | 69.7* | 31.3 | 30.8 | 18.6 | 20.1 | 14.8 | 20.2 | 5.0 | 5.8 | 4.5 |
| Ours | ∼15 | 1 | 93.4* | **66.9** | **66.6** | **41.6** | **46.4** | **40.0** | **45.0** | **39.7** | **37.2** | **32.2** |

## 3. Experiments

We consider the following datasets for training, including a widely used object detection dataset MS-COCO (Lin et al., 2014) and ImageNet training set (Deng et al., 2009). We consider some public naturally trained networks, *i.e.*, Inception-v3 (Inc-v3) (Szegedy et al., 2016), Inception-v4 (Inc-v4) (Szegedy et al., 2017), Resnet-v2-152 (Res-152) (He et al., 2016) and Inception-Resnet-v2 (IncRes-v2) (Szegedy et al., 2017), which are widely used for evaluating transferability. Besides, we supplement DenseNet-201 (Dense-201) (Huang et al., 2017), GoogleNet (Szegedy et al., 2015) and VGG-16 (Simonyan & Zisserman, 2014) and adversarially trained networks (Tramèr et al., 2018), *e.g.*, ens3-adv-Inception-v3 (Inc-v3$_{ens3}$), ens4-adv-Inception-v3 (Inc-v3$_{ens4}$) and ens-adv-Inception-ResNet-v2 (IncRes-v2$_{ens}$).

**Implementation details.** We choose the same ResNet autoencoder architecture in (Johnson et al., 2016; Naseer et al., 2019) as the basic generator networks, which consists of downsampling, residual and upsampling layers. Smoothing mechanism is proposed to improve the transferability against adversarially trained models (Dong et al., 2019). Instead of adopting smoothing for generated perturbation while the training is completed as CD-AP (Naseer et al., 2019), we introduce adaptive Gaussian smoothing kernel to compute $\delta$ from Eq. (2) in the training phase, named **adaptive Gaussian smoothing**, with the focus of making generated results obtain adaptive ability. More implementation details are illustrated in Appendix.

### 3.1. Transferability Evaluation

**Efficiency and effectivenessof multi-target black-box attack.** Among comparable methods, instance-specific methods, *i.e.*, MIM, TI-DIM, DIM and TI-DIM, require iterative mechanism with $M$ steps by computing gradients to obtain adversarial examples. Instance-agnostic methods only require the inference cost from the trained generator, thus possessing the priority for those attack scenarios within limited time. However, instance-specific methods require to train 8 models to obtain all predictions from 8 different classes. As a comparison, our conditional generative method only trains one model to inference the results and outperforms in the

*Table 2.* Transferability comparison on NeurIPS 2017 competition following the standard protocol with 1,000 stochastic target classes.

| Targeted Black-box Attack in NeurIPS 2017 Competition | | | | | |
|---|---|---|---|---|---|
| Method | Inc-v4 | IncRes-v2 | Res-152 | Dense-201 | GoogleNet | VGG-16 |
| MIM | 0.1 | <0.1 | <0.1 | 0.3 | 0.1 | <0.1 |
| TI-MIM | 0.2 | <0.1 | <0.1 | 0.1 | 0.2 | 0.2 |
| SI-MIM | 0.6 | 0.6 | 0.1 | 0.4 | 0.3 | 0.1 |
| DIM | 1.5 | 1.0 | <0.1 | 0.6 | <0.1 | 0.5 |
| TI-DIM | 0.6 | 0.6 | <0.1 | 0.3 | 0.3 | 0.3 |
| SI-DIM | 1.9 | 1.3 | 0.5 | 1.3 | 1.0 | 0.7 |
| Ours | 35.9 | 37.4 | 25.0 | 26.8 | 25.8 | 26.6 |

aspect of *efficiency*. Tab. 1 shows the transferability comparison of different methods on both naturally and adversarially trained models. The success rate of instance-specific attacks are lower than 3%. The instance-agnostic attack CD-AP obtains acceptable performance, yet inferior to proposed method w.r.t black-box transferability. The **primary reason** for such a trend lies in some distinctions as 1) direct clip projection in CD-AP and our smooth projection in Eq. (2) and 2) their Gaussian Smoothing and our adaptive Gaussian Smoothing. Thus proposed conditional generative method can be a reliable baseline w.r.t targeted black-box attacks.

**Effectiveness on NeurIPS 2017 Competition** We here follow the official setting from NeurIPS 2017 adversarial competition (Kurakin et al., 2018) for testing targeted black-box transferability. Considering limited resource, we only focus on the instance-specific attack. Our hierarchical partition mechanism considers 20 models, with each specifying 50 diverse classes from k-DPP hierarchical partition in this setting, to implement targeted attack by only once inference for each target image. Our method outperforms all other baseline methods in Tab. 2. The results demonstrate that this method can be reliable in practical targeted attacks, regarding both *effectiveness* and *efficiency*.

## 4. Discussion and Conclusion

Transferability of targeted black-box attack is affected by data and model. Instance-specific methods obtain weak transferability due to easily overfitting the data point and white-box model. Proposed generative method reduces the dependency for data points by learning from the unlabeled training data, thus enabling to learn semantic pattern and significantly improve the transferability of targeted attacks.

## A. Sampling Algorithm

We summarize the overall sampling procedure based on k-DPP (Kulesza & Taskar, 2011) as follows.

- Compute the RBF kernel matrix $L$ of $\phi_{cls}$ and eigen-decomposition of $L$.

- A random subset $V$ of the eigenvectors is chosen by regarding the eigenvalues as sampling probability.

- Select a new class $c_i$ to add to the set and update $V$ in a manner that de-emphaseizes items similar to the one selected.

- Update $V$ by Gram-Schmidt orthogonalization, and the distribution shifts to avoid points near those already chosen.

By performing the Algorithm 1, we can obtain a subset with $k$ size. Thus while handling the conditional classes with K, we can hierarchically adopt this algorithm to get the final $K/k$ subsets, which are regarded as conditional variables of generative models to craft adversarial examples.

## B. Some Implementation Details

**The study of smoothing mechanism.** Smoothing mechanism has been proved to improve the transferability against adversarially trained models. CD-AP (Naseer et al., 2019) uses direct clip projection to have a fixed norm $\epsilon$, and adopts smoothing for generated perturbation while the generator $\mathcal{G}$ is trained, *i.e.*,

$$\text{Train: } \boldsymbol{x}^*_{s_i} = \text{Clip}_\epsilon(\mathcal{G}(\boldsymbol{x}_{s_i}),$$
$$\text{Test: } \boldsymbol{x}^*_{s_i} = \text{W} * \text{Clip}_\epsilon(\mathcal{G}(\boldsymbol{x}_{s_i}), \qquad (4)$$

where W indicates Gaussian smoothing of kernel size of 3, $*$ indicates the convolution operation, and $\text{Clip}_\epsilon$ means clipping values outside the fixed norm $\epsilon$. As a comparison, we introduce adaptive Gaussian smoothing kernel to compute adversarial images $\boldsymbol{x}^*_{s_i}$ from in the training phase, named **adaptive Gaussian smoothing** as

$$\text{Train \& Test: } \boldsymbol{x}^*_{s_i} = \epsilon \cdot \text{W} * \tanh(\mathcal{G}(\boldsymbol{x}_{s_i}) + \boldsymbol{x}_{s_i}, \quad (5)$$

which can make generated results obtain adaptive ability in the training phase. We perform training in ImageNet dataset to report all results including comparable baselines.

**Network architecture of generator.** We adopt the same autoencoder architecture in (Naseer et al., 2019) as the basic generator networks. Besides, we also explore Big-GAN (Brock et al., 2018) as conditional generator network. An very weak testing performance is obtained even in the

**Algorithm 1** Sampling Algorithm by kDPP

**Input:** Weight Vector $\boldsymbol{\theta}_{cls}$; Subset size $k$.
**Output:** A subset $C$.
Compute RBF kernel matrix $L$ of $\boldsymbol{\theta}_{cls}$
Compute eigenvector/value $\{v_n, \lambda_n\}_{n=1}^N$ pairs of $L$
*// Phase I:*
$J \leftarrow \phi, e_k(\lambda_1, \ldots, \lambda_N) = \sum_{|J|=k} \prod_{n \in J} \lambda_n$
**for** n = N, ..., 1 **do**
   **if** $u \sim U[0,1] < \lambda_n \frac{e_{k-1}^{n-1}}{e_k^n}$ and $k > 0$ **then**
      $J \leftarrow J \cup \{n\}; k \leftarrow k - 1$
   **end if**
**end for**
*//Phase II:*
$V \leftarrow \{v_n\}_{n \in J}, Y \leftarrow \phi$
**while** $|V| > 0$ **do**
   Select $c_i$ from $\mathcal{C}$ with $\text{P}(c_i) = \frac{1}{|V|} \sum_{v \in V} (v^\top e_i)^2$
   $C \leftarrow C \cup \{c_i\}$
   $V \leftarrow V_\perp$, an orthonormal basis for the subspace of $V$
   orthogonal to $e_i$
**end while**

*white-box* attack scenario, possibly explained by the weak diversity of latent variable with the Gaussian distribution from BigGAN in the training phase, whereas autoencoder can take full advantage of large-scale training dataset, *e.g.*, ImageNet. Furthermore, we also train the autoencoder with Gaussian noise as the training dataset and obtain similar inferior performance in the white-box attack scenario, indicating that a large-scale training dataset is very significant for generating transferable targeted adversarial examples.

**Some details.** In our experiments of testing time, we apply NVIDIA 1080Ti GPUs. Instance-specific methods, *i.e.*, MIM, TI-DIM, DIM and TI-DIM, adopt iterative steps $M = 20$ and follow their reported hyperparameters.

## C. More Analyses

Targeted adversarial samples from proposed generative method can produce semantic pattern inherent to the target class in Fig. 3. Why does generative semantic pattern work?

First, *generative methods can produce strong targeted semantic pattern that is robust to the influence of data*, which is obtained by minimizing the loss of specific target class in the training phase. To corroborate our claim, we directly feed scaled crafted perturbations by instance-specific attack MIM and our generative method into the classifier. Indeed, we find that our generative perturbation is considered as target class with a high degree of confidence whereas the perturbation from MIM performs like the noise.

Second, *the generated adversarial semantic pattern*

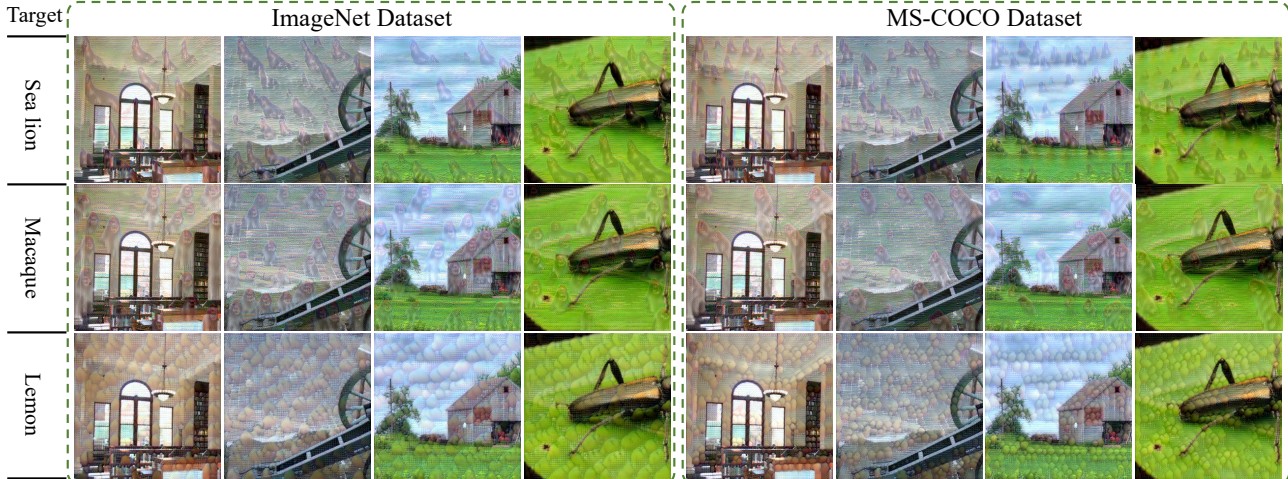

Figure 3. Generative examples of adversarial images with perturbation budget of $\ell_\infty \leq 16$. We separately adopt the ImageNet and MS-COCO dataset as the training dataset to implement the generation of targeted perturbations. Our method can generate semantic pattern independent of training dataset.

*achieves well-generalizing performance among the different models*. We feed 1k images from ImageNet test set into the generator trained by Inc-v3 model to obtain 1k semantic patterns, which are scaled to image pixel space and then fed into different classifiers. We compute the mean confidence of **0.46** for Dense-201, **0.44** for Inc-v4, and **0.35** for Res-152, whereas the perturbation from MIM is lower than $0.01$. The results show that our scaled semantic pattern can directly achieve well-generalizing performance among models, possibly explained by utilizing similar feature knowledge from the same class on different classifiers trained on same training data distribution. Thus similar pattern can be instrumental for transferability among models.

Therefore, transferability of targeted black-box attack is simultaneously affected by data and model. Instance-specific methods easily overfit the data point and white-box model, resulting in weak transferability. As a comparison, proposed generative method with powerful learning capacity reduces the dependency for data point by adopting the unlabeled training data, thus enabling the model to learn semantic pattern and improve the transferability of targeted black-box attack. We hope that crafting C-GSP can be regarded as a new reliable baseline method in terms of targeted black-box attacks, which raises new security issues to develop more robust DNNs.

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
