# OpenReview forum: "Boosting Transferability of Targeted Adversarial Examples via Hierarchical Generative Networks"
_ICML.cc/2021/Workshop/AML — ICML 2021 Workshop AML Poster_

### Official Review · Reviewer_t82W · 2021-06-20
**Review for generative targeted attack**

**Rating:** Accept
**Confidence:** 5

**Review:**

This paper analyzes targeted blackbox attack using generative models. The method utilizes a endocer-decoder network while combining a class-conditioned vector with the encoder feature as the latent vector. Authors also adapt two methods to improve the transferability. Plenty of experiments are conducted against both natual trained models and adversarially trained models. The proposed method outperforms baseline method by a significant margin.

There are some quesitions unclear for me:
1. What is the motivation to design such a model?
2. The ablation study should be conducted to demonstrate the effect of the two methods.

In summary, this paper propose an effective method to improve targeted black-box attack. It would make contributions to the workshop.

---

### Decision · Program_Chairs · 2021-06-21

**Decision:**

Accept (Poster)

**Comment:**

This paper proposed a targeted black-box attack method with better transferability. The improvements are significant.